# Career Capital and Well-Being: A Configurational Perspective

**DOI:** 10.3390/ijerph191610196

**Published:** 2022-08-17

**Authors:** Qian Xu, Zhe Hou, Chao Zhang, Feng Yu, Tong Li

**Affiliations:** 1School of Education Science, Shanxi Normal University, Taiyuan 030000, China; 2School of Philosophy, Wuhan University, Wuhan 430072, China

**Keywords:** human capital, social capital, psychological capital, well-being, fuzzy-set qualitative comparative analysis

## Abstract

This study explored the configuration effect of human capital, social capital, and psychological capital on employee well-being. A total of 458 employees were investigated via a human capital scale, social capital scale, Chinese psychological capital scale, and multiple well-being questionnaire. The result of the fuzzy-set qualitative comparative analysis showed that human capital, social capital, and psychological capital in the form of diversified configuration will achieve high well-being, characterized by “all roads lead to Rome”. Even without human capital and social capital, high well-being can be achieved as long as psychological capital exists. Psychological capital is the most critical factor affecting subjective well-being, followed by human capital and social capital. Compared with guanxi-oriented psychological capital, task-oriented psychological capital is a more critical factor in achieving high well-being.

## 1. Introduction

In recent years, with the vigorous development of well-being-oriented human resource management [1], the positive impact of well-being on employee turnover intention [2], job performance [3], career success [4], and other variables has received extensive attention. Therefore, improving employees’ well-being has become an important goal of enterprise human resource management [5]. Based on the philosophical basis of hedonism and realism, well-being can be divided into subjective and psychological well-being [6]. Subjective well-being emphasizes life satisfaction at the cognitive level and positive emotions at the emotional level. Psychological well-being emphasizes the positive psychological experience brought by self-realization and the meaning of life [6].

In the era of boundaryless careers, an intelligent career is crucial to individual career development. Intelligent career theory proposes three employabilities for individuals to adapt to enterprise development. These three employabilities, which can be collectively referred to as career capital [7], represent three types of capital for individuals, respectively: human capital, social capital, and psychological capital. Career capital reflects individual competitiveness in the workplace [8]. Human capital, social capital, and psychological capital emphasize knowledge and skills [9], social networks and resources [10], and positive psychological quality [11], respectively.

Career capital is beneficial to individual career development and well-being. Xu et al. [12] found that psychological capital had the most significant positive effect on employees’ subjective well-being and psychological well-being, followed by social capital and human capital. Xu et al. [13], based on the person-centered perspective, through latent profile analysis, divided well-being into “happy type”, “general type”, and “painful type”. The study found that human capital mainly affects the well-being of the general type [13]. Compared with social capital, mobilized social capital was a more important factor affecting well-being [13]. Psychological capital mainly affects the well-being of the general type and the happy type [13].

Although previous studies adopted the research perspectives of “variable-centered” and “person-centered” to investigate the impact of the three types of capitals on employees’ well-being [12,13], they were all based on multiple regression analysis. They all calculated the marginal “net effect” among variables [14], reflecting the symmetric variable relationship. However, there is a complex interaction among human capital, social capital, and psychological capital which does not conform to the assumption of multiple regression. Addressing the research gap, it will help enrich the understanding of the relationship among the three types of career capital in the intelligent career theory, and help enterprises and employees make more targeted investments in the three types of career capital.

Human capital, social capital, and psychological capital may affect well-being in a configuration way because of the interdependence among the three types of career capital. The configuration effect means that fuzzy-set qualitative comparative analysis (fsQCA) adopts a holistic perspective to carry out case-oriented comparative analysis and each case is regarded as the “configuration” of conditional variables [14]. The purpose of fsQCA analysis is to find out the causal relationship between the condition configuration and the result by comparing the cases, and answering the question of “which configuration of the condition can lead to the expected result?” [14] When a study conducts a holistic analysis of the condition configuration, it has adopted the assumption that the conditions are inter-dependent, which is more in line with the social phenomenon [14].

Therefore, considering the interaction among the three capitals, this study adopted fsQCA to investigate the following problems: (1) What configurations exist among the three types of career capital to achieve well-being? (2) What are the characteristics of each configuration, and what causal asymmetry do they embody?

## 2. Theoretical Background

### 2.1. Well-Being

As an essential issue in career development, well-being significantly impacts individual physical and mental health and job performance [15]. According to the philosophical basis of hedonism and realism, well-being is usually divided into subjective well-being and psychological well-being [16]. Subjective well-being refers to an individual’s assessment of his overall life satisfaction and happiness [17], which is generally divided into three dimensions: life satisfaction, positive emotions, and negative emotions [18,19]. Psychological well-being refers to the psychological experience brought about by self-actualization [20], which emphasizes that individuals perceive the meaning of their existence through activities and feel happy after reaching their goals in the process of realizing their self-value. Psychological well-being includes six dimensions: self-acceptance, positive relations with others, autonomy, environmental mastery, purpose in life, and personal growth [21].

### 2.2. Career Capital and Well-Being

#### 2.2.1. Human Capital and Well-Being

Human capital emphasizes that human beings, as a kind of capital, have more significant appreciation potential than currency and other substances. Human capital is the opposite of “physical capital” and reflects the total stock of production knowledge, labor and management skills, and health quality contained in human beings [9]. Many studies have proved that human capital has positive effects on well-being. Florida et al. [22] found that human capital played a positive role in the formation of urban happiness, surpassing the influence of income and other variables. Layard [23] confirmed that people’s education level indirectly improved well-being by increasing income. Liu [24] found that education level was significantly positively correlated with employees’ emotional happiness. At the same time, human capital had a more significant impact on well-being at the overall level (e.g., country) than at the individual level [25].

However, although many studies show that human capital has a significant positive predictive effect on well-being, the relationship between human capital and well-being is still controversial. Layard and Layard [23] found that although education could effectively improve people’s income, it could not significantly improve subjective well-being. Liu [24] found no significant difference in job satisfaction among employees with different education levels. Xu et al. [13] found that implicit human capital could negatively predict subjective well-being and the psychological well-being of employees with average well-being.

#### 2.2.2. Social Capital and Well-Being

Social capital refers to social resources such as individual, interpersonal communication, work contact network, and mutual trust [10]. Social capital includes accessed and mobilized social capital [26]. The former refers to the status of parents and the breadth of social connections, which determines the scope of accessed social resources [26]. The latter emphasizes social resources, including the status of the used acquaintances and the strength of their contacts [26]. There are many ways to measure social capital, including social network analysis, peer nomination, job generation, and resource generation. 

According to social interdependence theory, positive relationships can promote positive interactions, the realization of individual goals, and the well-being of individuals [27]. Many scholars have generally recognized that social capital positively promotes residents’ well-being, and that there was a long-term stable co-integration relationship between social capital and subjective well-being [28,29]. A meta-analysis by Chu et al. [30] showed that social capital positively affected well-being. However, the relationships between social capital and happiness were different due to different types of social support and measurement methods [30]. 

#### 2.2.3. Psychological Capital and Well-Being

Psychological capital is a measurable, developable, and positive psychological ability that can promote job performance. Luthans et al. [11] explored the four dimensions of psychological capital: confidence, optimism, resilience, and hope. Later, Ke et al. [31] compiled the Chinese psychological capital scale, which broadened the meaning of psychological capital and divided it into task-oriented psychological capital and guanxi-oriented psychological capital. Task-oriented psychological capital includes self-confidence, courage, optimistic, hope, enterprise, and tenaciousness, while guanxi-oriented psychological capital includes tolerance, forgiveness, respect, comity, modesty, honesty, gratitude, and dedication. In fact, task-oriented psychological capital is Luthans’ psychological capital. Therefore, compared with the Western concept of psychological capital, Chinese psychological capital emphasizes more social psychological capital, which is consistent with Chinese collectivist culture.

Psychological capital can effectively stimulate individuals’ positive emotions, help them cope effectively with changes and challenges at work, cultivate a positive work attitude, develop emotional commitment at a higher level, and promote individuals’ well-being [32]. Researchers have demonstrated that well-being is positively affected by hope [33], resilience [34], self-efficacy [35], and optimism [36].

### 2.3. Configurational Relationships among Three Types of Career Capital

Previous studies have examined the marginal net effect of three types of career capital on well-being and the symmetric variable relationship through multiple regression [12,13,14]. However, multiple regression has some defects, including the relationship among human capital, social capital, and psychological capital not conforming to the assumptions of multiple regression.

Firstly, multiple regression assumes that variables are independent; that is, the effects of human capital, social capital, and psychological capital on well-being are independent. However, the relationship among the three types of capital does not conform to this assumption. Specifically, there is an intersection in the conceptual meaning of the three capitals. Human capital and social capital contain moral elements [37]; both social capital and Chinese psychological capital contain interpersonal relationship elements [31]. Moreover, the three capitals affect well-being through extremely complex interactions [38]; that is, there is a strong correlation among the three capitals and their impact on well-being has a configuration effect. Therefore, traditional multiple regression simplifies the complex relationship among the three capitals, resulting in statistical tests that do not conform to the premise hypothesis. Even though the structural equation model can examine the correlation among the three capitals, the essence of the calculated path coefficient is still the marginal “net effect” [14].

Secondly, multiple regression assumes causal symmetry among variables [14]. For example, based on multiple regression, Xu et al. [12] found that high well-being was caused by high human capital and low well-being was caused by low human capital. However, human capital (such as educational level) may not be related to well-being [24] and may even negatively predict well-being [13]. Low human capital can both achieve low well-being and high well-being. This inconsistent result was affected by the asymmetric relationships and interdependence among variables (i.e., configuration). Due to the highly complex relationship among the three capitals, the interdependence among the three may produce causal asymmetry among variables.

FsQCA assumes that independent variables are interdependent, and that there is a causal asymmetry between variables that can overcome the above-mentioned defects of multiple regression [14]. Therefore, this study used fsQCA to investigate the configuration effect of human capital, social capital, and psychological capital on well-being due to their interdependence and the asymmetric causality among them.

## 3. Materials and Methods

### 3.1. Participants

The participants were selected by convenient sampling and came from 26 enterprises in nine provinces of China, involved in 13 industries such as aquaculture, new energy, and food processing. Participants completed four Likert scales by self-report. This study set some anti-counterfeiting items, such as “Please directly select ‘fully agree’.” The valid data are the data of the participant who answers these items correctly. Among 648 distributed questionnaires, 458 valid questionnaires were recovered, with an effective rate of 70.68%.

### 3.2. Materials

#### 3.2.1. Human Capital Questionnaire

The human capital scale compiled by Ke et al. [39] used a single factor. The scale had three items (e.g., “I have more work experience than my colleagues”). Participants were scored by a Likert 6-point scale (1 = strongly disagree, 6 = strongly agree). Cronbach’s α of the scale was 0.95.

#### 3.2.2. Social Capital Scale

The social capital scale compiled by Wang [40] included accessed social capital and mobilized social capital. The scale had 23 items (e.g., “I have a large circle of friends”). Participants were scored by a Likert 6-point scale (1 = strongly disagree, 6 = strongly agree). The model fitted well (χ^2^/*df* = 3.53, TLI = 0.92, CFI = 0.93, RMSEA = 0.07, RMR = 0.05), with Cronbach’s αs of total scale and each dimension of 0.95, 0.92, and 0.94, respectively.

#### 3.2.3. Chinese Psychological Capital Scale

The Chinese psychological capital scale compiled by Ke et al. [31] included task-oriented psychological capital and guanxi-oriented psychological capital. The scale had 40 items (e.g., “I have full confidence in my work ability”). Participants were scored by a Likert 6-point scale (1 = strongly disagree, 6 = strongly agree). The model fitted well (χ^2^/*df* = 2.31, TLI = 0.91, CFI = 0.92, RMSEA = 0.05, RMR = 0.05), with Cronbach’s αs of total scale and each dimension of 0.96, 0.93, and 0.94, respectively.

#### 3.2.4. Multiple Well-Being Questionnaire

The multiple well-being questionnaire compiled by Miao [6] included subjective well-being and psychological well-being. The questionnaire had 50 items (e.g., “I am satisfied with my life”). Participants were scored by a Likert 7-point scale (1 = strongly disagree, 7 = strongly agree). The model fitted well (χ^2^/*df* = 2.33, TLI = 0.92, CFI = 0.92, RMSEA = 0.05, RMR = 0.05), with Cronbach’s αs of total scale and each dimension of 0.96, 0.86, and 0.97, respectively.

### 3.3. Statistical Analysis

Human capital, social capital, and psychological capital are interdependent and have a phenomenon of “All roads lead to Rome”, which does not accord with the hypothesis of multiple regression. Therefore, this study used the configuration perspective to examine the relationship between the three types of career capital and well-being. The main methods used in fsQCA are Boolean algebra and set analysis, which assumes that the independent variables are interdependent and work together, and there is no optimal. Moreover, fsQCA believes that there are equivalent multiple paths or solutions [14,41].

Because the Likert scale data do not meet the conditions of Boolean logic analysis, the score needs to be calibrated through fsQCA 3.1 to convert the original data into the set data between 0 and 1. The human capital scale, social capital scale, and psychological capital scale adopt Likert 6-point scoring, so “1” was defined as complete non-membership, “3.5” was the maximum fuzzy point, and “6” was complete membership. The comprehensive well-being questionnaire adopted Likert 7-point scoring, so “1” was defined as complete non-membership, “4” was the maximum fuzzy point, and “7” was complete membership. Through the setting of three thresholds, original scores were converted into a fuzzy membership degree between 0 and 1. Fsqca3.1 software was used to analyze the necessity of antecedent conditions and the standard of antecedent configuration.

## 4. Results

### 4.1. Necessity Analysis

The necessity of each antecedent condition to well-being was analyzed, as shown in Table 1. When the outcome variable was subjective well-being, the consistency of mobilized social capital, task-oriented psychological capital, and guanxi-oriented psychological capital was 0.91, 0.99, and 0.99, respectively, all greater than 0.90. Therefore, mobilized social capital, task-oriented psychological capital, and guanxi-oriented psychological capital were necessary to achieve subjective well-being.

When the outcome variable was psychological well-being, the consistency of task-oriented psychological capital and guanxi-oriented psychological capital was 0.98 and 0.91, respectively, both greater than 0.90. This indicated that task-oriented psychological capital and guanxi-oriented psychological capital were necessary conditions to achieve psychological well-being.

### 4.2. Configurations for Well-Being

The antecedent conditions were incorporated into the standard analysis to analyze the configuration solutions affecting well-being. The sample size was 458, so the frequency threshold was set as 3 [41,42]. The intermediate solutions in the standard analysis are shown in Table 2 and Table 3.

The results in Table 2 showed that the antecedent configuration of subjective well-being included four combinations, with an overall consistency of 0.75 and overall coverage of 0.75, indicating that the overall configuration had a strong interpretation of the results. The four configurations were, respectively, “~ accessed social capital * task-oriented psychological capital * guanxi-oriented psychological capital” (configuration A); “ ~ mobilized social capital * task-oriented psychological capital * guanxi-oriented psychological capital” (configuration B); “human capital * ~ accessed social capital * ~ mobilized social capital * task-oriented psychological capital” (configuration C); and “ ~ human capital * task-oriented psychological capital * guanxi-oriented psychological capital” (configuration D). The consistencies of the four configurations were 0.81, 0.80, 0.88, and 0.81, respectively. This indicates that when the configuration existed, the cases were 81%, 80%, 88%, and 81% likely to achieve subjective well-being, respectively. The raw coverage of the four configurations was 0.65, 0.62, 0.53, and 0.57, respectively, indicating that the four configurations explained 65%, 62%, 53%, and 57% of the cases, respectively. Configuration D’s raw and unique coverages were the smallest, so configuration D was a relatively minor factor in achieving subjective well-being.

The results in Table 3 showed that the antecedent configurations of psychological well-being included three combinations, with the overall consistency of 0.98 and overall coverage of 0.83, indicating that the antecedent configurations had a strong explanatory power to the results. The three configurations were “accessed social capital * task-oriented psychological capital * guanxi-oriented psychological capital” (configuration E); “mobilized social capital * task-oriented psychological capital * guanxi-oriented psychological capital” (configuration F); and “human capital * ~ accessed social capital * ~ mobilized social capital * task-oriented psychological capital” (configuration G). The consistency of the three configurations was 0.99, indicating that when this configuration existed, cases were 99% likely to achieve high psychological well-being. The raw coverage of the three configurations were 0.73, 0.73, and 0.37, respectively, indicating that the three configurations explained 73%, 73%, and 37% of the cases, respectively. The raw coverage of configuration E was equal to that of configuration F, while the unique coverage of configuration F was slightly larger than that of configuration E. Therefore, the effect of configuration F on psychological well-being was slightly larger than configuration E’s effect. Configuration G’s raw coverage and unique coverages were the smallest, so configuration G was a relatively minor factor in achieving psychological well-being.

## 5. Discussion

### 5.1. Configurations to Achieve High Well-Being

Task-oriented psychological capital and guanxi-oriented psychological capital were the necessary conditions for achieving high subjective and high psychological well-being, indicating that psychological capital was an essential factor affecting well-being, consistent with Xu et al. [12]. Mobilized social capital was a necessary condition to achieve high subjective well-being but not a necessary condition to achieve high psychological well-being, indicating that mobilized social capital mainly affected subjective well-being. Although the mobilized social capital and guanxi-oriented psychological capital were necessary to achieve high well-being, their consistency did not reach 1, and they still did not exist in the antecedent configuration of achieving high well-being.

There are similarities between the antecedent configurations of high subjective and psychological well-being. Firstly, there is the same configuration to achieve high subjective and psychological well-being: configurations C and G “human capital * ~ accessed social capital * ~ mobilized social capital * task-oriented psychological capital”. Moreover, configurations C and G’s raw coverage and unique coverage were the most minor, indicating a few cases of lack of social capital. Configurations C and G showed that when social capital and guanxi-oriented psychological capital were lacking, human capital and task-oriented psychological capital were needed to achieve high well-being. Secondly, in addition to configurations C and G, task-oriented and guanxi-oriented psychological capital existed in other configurations. It shows that in the cases of achieving high well-being, most cases have task-oriented and guanxi-oriented psychological capital at the same time, reflecting the important role of psychological capital in achieving high well-being. Moreover, task-oriented psychological capital exists in all configurations to achieve high well-being, while guanxi-oriented psychological capital is lacking in configurations C and G. Therefore, task-oriented psychological capital is the most important factor in achieving high well-being. However, Xu et al. [13] found that only guanxi-oriented psychological capital can affect happy-type employees’ well-being. It shows that task-oriented psychological capital is the most critical factor in achieving high well-being when considering the interdependence among the three capitals. When evaluating the non-overlapping effect between variables, guanxi-oriented psychological capital is a critical factor in achieving high well-being.

In the antecedent configuration of achieving high subjective well-being, task-oriented psychological capital appeared four times, guanxi-oriented psychological capital appeared three times, human capital appeared once, and both kinds of social capital appeared zero times. It indicates that psychological capital is the most critical factor for subjective well-being, followed by human and social capital. Xu et al. [12] found that social capital had a more significant impact on subjective well-being than human capital. This inconsistent result was related to whether to consider the interdependence among the three capitals. Configurations A, B, and D show that when human capital and social capital do not exist, the existence of both kinds of psychological capital can also achieve high subjective well-being. On the one hand, it reflects that psychological capital is the most critical factor affecting subjective well-being. On the other hand, it reflects the asymmetric causal relationship between the three types of capital and subjective well-being.

In the antecedent configuration of achieving high psychological well-being, task-oriented psychological capital appeared three times, guanxi-oriented psychological capital appeared twice, social capital, mobilized social capital, and human capital appeared once. The configuration of human capital had the lowest coverage. It indicated that psychological capital is the most critical factor affecting psychological well-being, followed by social and human capital. It is consistent with the result of Xu et al. [12]. Two kinds of psychological capital existed in configurations A, B, E, and F, but the accessed social capital did not exist in configurations A and E, and the mobilized social capital did not exist in B and F. It shows that compared with high subjective well-being, the realization of high psychological well-being depends more on social capital. Hence, the impact of social capital on psychological well-being is greater than human capital. Comparing configurations E, F, and G, the role of guanxi-oriented psychological capital on psychological well-being depends on the existence of social capital, and the role of task-oriented psychological capital on psychological well-being cannot rely on social capital. That is, the existence of social capital is the premise of guanxi-oriented psychological capital to achieve high psychological well-being, but not the premise of task-oriented psychological capital.

Compared with the international concept of psychological capital, Chinese psychological capital emphasizes more social psychological capital, such as tolerance, forgiveness, respect, comity, modesty, honesty, gratitude, and dedication. Thus, international academic circles pay more attention to task-oriented psychological capital and neglect guanxi-oriented psychological capital. In this study, compared with guanxi-oriented psychological capital, task-oriented psychological capital is a more critical factor in achieving high well-being. Therefore, even in a collectivist society, task-oriented psychological capital is still more important.

Previous studies used regression analysis, and there is great controversy about the relationship between human capital and well-being. By examining the asymmetric configuration effect among variables, when achieving a high level of well-being, human capital can be high, non-high, and both high and non-high. It is similar to the positive, negative, and uncorrelated results found in previous studies, respectively. Therefore, from the perspective of regression analysis, the inconsistent results of positive correlation, negative correlation, and uncorrelation between human capital and well-being are not conflicting and contradictory from the perspective of asymmetry.

### 5.2. Management Implications

In management practice, employers and employees should realize that when individuals lack human capital or social capital, as long as psychological capital exists, they can also obtain high well-being. Therefore, employers should focus on cultivating employees’ task-oriented psychological capital (i.e., self-confidence, courage, enterprise, optimism, hope, and resilience) and guanxi-oriented psychological capital (i.e., humility, honesty, respect and comity, inclusiveness, forgiveness, gratitude, and dedication), especially task-oriented psychological capital. Social capital has a greater impact on subjective well-being than human capital, while human capital has a greater impact on psychological well-being than social capital, so social capital and human capital should be regarded as equally important. Moreover, the accessed social capital (i.e., social resources, network differences, network scale) and mobilized social capital (i.e., acquaintance support, family support, and friend support) should also be regarded as equally important.

### 5.3. Research Prospect

This study examined the configuration effect of human capital, social capital, and psychological capital on well-being. In the future, scholars can explore the impact of three capitals on career adaptability, career success, job performance, and other workplace outcome variables. Scholars can also compare the results of multiple regression and fsQCA. In addition, in the intelligent career theory, “knowing-why” is often regarded as psychological capital, which is actually not very appropriate. In this theory, “knowing-why” emphasizes motivation, belief, and value, and does not just emphasize psychological capital. Thus, there may be other capital forms in “knowing-why”. Scholars can explore other career capital in “knowing-why”.

## 6. Conclusions

There are four configurations to achieve high subjective well-being and three configurations to achieve high psychological well-being. “Human capital * ~ accessed social capital * ~ mobilized social capital * task-oriented psychological capital” is a common configuration to achieve high subjective and psychological well-being. Psychological capital is the most critical factor for well-being. Human capital is a more important factor for subjective well-being than social capital, while social capital is a more important factor for psychological well-being than human capital. The realization of high psychological well-being depends more on social capital. Compared with guanxi-oriented psychological capital, task-oriented psychological capital is a more critical factor in achieving high well-being. The effect of guanxi-oriented psychological capital on psychological well-being depends on social capital.

## Figures and Tables

**Table 1 ijerph-19-10196-t001:** Necessity analysis of antecedent conditions for well-being.

Antecedent Condition	Subjective Well-Being		Psychological Well-Being	
	Consistency	Coverage	Consistency	Coverage
Human capital	0.89	0.69	0.78	0.97
~ Human capital	0.58	0.78	0.43	0.95
Accessed social capital	0.89	0.74	0.74	0.99
~ Accessed social capital	0.65	0.79	0.49	0.96
Mobilized social capital	0.91	0.74	0.74	0.99
~ Mobilized social capital	0.63	0.77	0.48	0.95
Task-oriented psychological capital	0.99	0.57	0.98	0.91
~ Task-oriented psychological capital	0.27	0.95	0.17	0.99
Guanxi-oriented psychological capital	0.99	0.61	0.95	0.94
~ Guanxi-oriented psychological capital	0.38	0.95	0.25	0.99

Note. The symbol “~” indicates that the condition does not exist.

**Table 2 ijerph-19-10196-t002:** Configurations for achieving high subjective well-being.

Number	Configuration	Raw Coverage	Unique Coverage	Consistency	Overall Solution Coverage	Overall Solution Consistency
A	~ Accessed social capital * task-oriented psychological capital * guanxi-oriented psychological capital	0.65	0.03	0.81	0.75	0.75
B	~ Mobilized social capital * task-oriented psychological capital * guanxi-oriented psychological capital	0.62	0.03	0.80		
C	Human capital * ~ accessed social capital * ~ mobilized social capital * task-oriented psychological capital	0.53	<0.01	0.88		
D	~ Human capital * task-oriented psychological capital * guanxi-oriented psychological capital	0.57	0.05	0.81		

Note. The symbol “~” indicates that the condition does not exist and “*” indicates the logic “and”.

**Table 3 ijerph-19-10196-t003:** Configurations for achieving high psychological well-being.

Number	Configuration	Raw Coverage	Unique Coverage	Consistency	Overall Solution Coverage	Overall Solution Consistency
E	Accessed social capital * task-oriented psychological capital * guanxi-oriented psychological capital	0.73	0.04	0.99	0.83	0.98
F	Mobilized social capital * task-oriented psychological capital * guanxi-oriented psychological capital	0.73	0.05	0.99		
G	Human capital * ~ accessed social capital * ~ mobilized social capital * task-oriented psychological capital	0.37	0.04	0.99		

Note. The symbol “~” indicates that the condition does not exist and “*” indicates the logic “and”.

## Data Availability

Data will be provided by the authors on request.

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
