# Peer review of "Career Capital and Well-Being: A Configurational Perspective"

_ijerph, 2022, doi:10.3390/ijerph191610196_

Round 1
Reviewer 1 Report
This is a very interesting issue related to human capital in the context of well-being. Such an issue is extremely important - especially if we are talking in science about the multidisciplinarity of science. I am therefore pleased that such analyses are being carried out.
Nevertheless, I see a number of significant errors and shortcomings.
Firstly - an in-depth survey of the world literature should be done.
Secondly, the aim should be clearly defined and research questions should be posed, which should be verified in conclusion.
Thirdly, there is a lack of recommendations for future research.
Fourth - it would be useful to identify what research gap the article under investigation fills. What does it contribute to science, especially social science.
Fifth - I do not fully understand the idea of guanxi-capital - please clearly define this concept.
Sixth - please clearly state how the research group was selected, what methods were used, whether it is representative.
Seventh - it is worth pointing out what new the article brings to research both in China and internationally. What do we learn new through it?
Eight - it is necessary to detail the conclusions
Author Response
Dear reviewer:
We thank Reviewer 1 for these positive comments. We believe that we have been able to address your comments and suggestions and that our paper has been substantially improved as a result. According to the great suggestions, we followed these suggestions and made subsequent changes to the manuscript by using the ‘Track Changes’. Below, we indicate how we responded to each of your comments.
Comment 1: An in-depth survey of the world literature should be done.
Response to comment 1: We searched for more literature based on the reviewer’s comment. Especially the introduction of fsQCA. We have added more information about this statistical method in the article. As for the relationship between the three types of career capital and well-being, we think that our current literature review has reflected the current situation.
Comment 2: The aim should be clearly defined and research questions should be posed, which should be verified in the conclusion.
Response to comment 2: Actually, we wrote the study’s aim and research questions at the end of the introduction. ‘Considering the interaction among three capitals, this study adopted fuzzy-set qualitative comparative analysis (fsQCA) to investigate the following problems: (1) What configurations exist among the three types of career capital to achieve well-being? (2) What are the characteristics of each configuration, and what causal asymmetry does it embody?’ We have marked this part in red font and explained them in the conclusion.
Comment 3: There is a lack of recommendations for future research.
Response to comment 3: Thank you for the reviewer’s advice, the recommendation for future research is very important. We added the new section ‘5.3. Research Prospect’ to discuss the recommendations for future research.
This study examined the configuration effect of human capital, social capital, and psychological capital on well-being. In the future, scholars can explore the impact of three capitals on career adaptability, career success, job performance, and other workplace outcome variables. And scholars can compare the results of multiple regression and fsQCA. In addition, in the intelligent career theory, ‘knowing-why’ is often regarded as psychological capital, which is not very appropriate. In this theory, ‘knowing-why’ emphasizes motivation, belief, and value, it does not just emphasize psychological capital. So there may be other capital forms in ‘knowing-why’. Scholars can explore other career capital in ‘knowing-why’.
Comment 4: It would be useful to identify what research gap the article under investigation fills. What does it contribute to science, especially social science.
Response to comment 4: It is very important to reveal the gap and contribution. We have added these contents to the manuscript.
This study is a supplement to Xu et al. (2021) and Xu et al. (2022), which assumed that the three types of career capital are independent of each other and only examined the symmetrical ‘net effect’ of three capitals on well-being. But, the three types of career capital influence each other, depend on each other and even have an asymmetric relationship with well-being. The three types of capital affect well-being in the form of configuration rather than net effect. The relationship between the three types of capital and well-being is causal asymmetric rather than symmetric. Therefore, there is no research to explore the impact of the three capital on happiness from the perspective of configuration, and the cause and effect asymmetry among them. So it is the research gap. Address, the gap will help enrich the understanding of the relationship among the three types of career capital in the intelligent career theory, and help enterprises and employees make more targeted investments in the three types of career capital. It is also the contribution.
Comment 5: I do not fully understand the idea of guanxi-capital - please clearly define this concept.
Response to comment 5: Sorry, we have not explained this in detail in the original manuscript. So we have added these contents to the manuscript. Task-oriented psychological capital includes self-confidence, courage, optimism, hope, enterprising, and tenacious, while guanxi-oriented psychological capital includes tolerance, forgiveness, respect, comity, modesty, honesty, gratitude, and dedication. In fact, task-oriented psychological capital is Luthans’ psychological capital. Therefore, compared with the western concept of psychological capital, Chinese psychological capital emphasizes more social psychological capital, which is consistent with Chinese collectivist culture.
Comment 6: Please clearly state how the research group was selected, what methods were used, whether it is representative.
Response to comment 6: We have added these contents to section '3.1. Participants'. The participants were selected by convenient sampling. Participants completed the Likert scale by self-report. This study set some anti-counterfeiting items, such as "Please directly select 'fully agree.'" The valid data is the data of the participant who answers these items correctly.
Comment 7: It is worth pointing out what new the article brings to research both in China and internationally. What do we learn new through it?
Response to comment 7: We thank the reviewer for this important comment, it is very important to reveal the research significance of the manuscript. We have added these contents to the Discuss.
As we mentioned above, compared with the international concept of psychological capital, Chinese psychological capital emphasizes more social psychological capital, such as tolerance, forgiveness, respect, comity, modesty, honesty, gratitude, and dedication. So, the international community pays more attention to task-oriented psychological capital and neglects guanxi-oriented psychological capital. In this study, compared with guanxi-oriented psychological capital, task-oriented psychological capital is a more critical factor in achieving high well-being. Therefore, even in a collectivist society, task-oriented psychological capital is still more important.
Comment 8: It is necessary to detail the conclusions.
Response to comment 8: We thank the reviewer for this important comment. We explained this in great detail in the discussion section '5.1. Configurations to Achieve High Well-Being'. Since another reviewer wanted us to summarise the main findings in the conclusion, we refined the conclusion section. Do you think this is OK?
Thank you for your consideration. I look forward to hearing from you.
Kind regards,
Qian Xu
Reviewer 2 Report
Thank you for the opportunity to review your paper. I described below some suggestions to improve the quality of your study:
Title: does not inform the research design, participants and data analysis.
Abstract: Methods should include data analysis
Results: should include values that inform what type of correlation was found.
Conclusion: the study needs to summarise the conclusion- main findings.
Page 2, line 50- Data analysis should be in your methods section. The same happens on Page 2, line 55.
Page 2- Line 67- The paragraph below needs to be rewritten not to generalise affirmation based on one research:
“People with high subjective well-being are more satisfied with life, often experience positive emotions, and rarely experience negative emotions. However, people with low subjective well-being are unsatisfied with their life and often experience negative and less positive 70 emotions” [19].
Objectives, please clarify (in the introduction) to readers what “configuration effects” means.
3. Materials and Methods- Please state your study design and why the design is the best according to your objectives.
3.1. Participants: inform inclusion/exclusion criteria. How and where did you find them, and how did you approach them?
3.2. Materials: please be more descriptive of your measurements, including how to apply them, how long it takes, and what the main results are expected (classifications, levels, if appropriate, etc.). Please, inform the reliability and validity of these measures by citing the respective studies.
There is no information on data collection: how long the research was available to participants, where this Survey was stored and how they completed the measures.
Discussion: please include in the debate that this type of research design identifies trends; however, due to the complexity of Career Capital and Well-Being subject, it is important to conduct future qualitative studies. Qualitative studies can give you more data about “meaning”, whereas quantitative will show you trends or relations. Moreover, it is essential to address in your discussion the influence of Chinese culture on work and well-being that might differ when compared with other countries and continents.
Author Response
Dear reviewer:
We thank Reviewer 2 for these important comments. We apologize that the prior version of this paper has some errors and shortcomings. It is very helpful to improve the quality of our manuscript. According to the great suggestions, we followed these suggestions and made subsequent changes to the manuscript by using the ‘Track Changes’.
Comment 1: Title does not inform the research design, participants and data analysis.
Response to comment 1: We thank the reviewer for this important comment. This title does not inform the research design, participants, and data analysis.
Comment 2: Abstract: Methods should include data analysis
Response to comment 2: We thank the reviewer for this important comment. We added an introduction to the method in the abstract. ‘The result of the fuzzy-set qualitative comparative analysis showed ...’
Comment 3: Results: should include values that inform what type of correlation was found.
Response to comment 3: We thank the reviewer for this important comment. FsQCA is a combination of qualitative and quantitative methods. The correlation is reflected by the consistency coefficient, which we have interpreted in the results section
Comment 4: Conclusion: the study needs to summarise the conclusion- main findings.
Response to comment 4: We thank the reviewer for this important comment. Our conclusion is indeed in some seemingly redundant sentences, such as ‘Even without human capital and social capital, high well-being can be achieved as long as psychological capital exists;’ and some repetitive sentences, such as ‘Psychological capital is the most critical factor for ...’ Therefore, to summarise the main findings, we revised the conclusion.
Comment 5: Page 2, line 50- Data analysis should be in your methods section. The same happens on Page 2, line 55.
Response to comment 5: We thank the reviewer for this important comment. We have supplemented this in detail in the section 'Statistical Analysis.'
Because human capital, social capital, and psychological capital are interdependent and have a phenomenon of 'All roads lead to Rome', which does not accord with the hypothesis of multiple regression. Therefore, this study used the configuration perspective to examine the relationship between the three types of career capital and well-being. The main methods used in fsQCA are Boolean algebra and set analysis, which assumes that the independent variables are interdependent and work together, and there is no optimal. Moreover, fsQCA believes that there are equivalent multiple paths or solutions (Rihoux & Ragin, 2009), so it is possible to explore the combination of variables from 'chemical reaction’.
However, since our research questions come from the misuse of previous statistical methods, we suggest that it is better to retain a brief reference to the methods in the introduction.
Comment 6: Page 2- Line 67- The paragraph below needs to be rewritten not to generalise affirmation based on one research:
“People with high subjective well-being are more satisfied with life, often experience positive emotions, and rarely experience negative emotions. However, people with low subjective well-being are unsatisfied with their life and often experience negative and less positive emotions” [19].
Response to comment 6: We thank the reviewer for this important comment. This sentence refers to the dimensions of subjective well-being. But we use redundant sentences to express it. Therefore, we delete this sentence.
Comment 7: Objectives, please clarify (in the introduction) to readers what ‘configuration effects’ means.
Response to comment 7: We thank the reviewer for this important comment. Configuration effect means that fsQCA adopts a holistic perspective to carry out case-oriented comparative analysis and each case is regarded as the ‘configuration’ of conditional variables (Rihoux & Ragin, 2009). The purpose of fsQCA analysis is to find out the asymmetric causality between the condition configuration and the result by comparing the cases, and answering the question of 'which configuration of the condition can lead to the expected result?' When a study conducts a holistic analysis of the condition configuration, it has adopted the assumption that the conditions are inter-dependent, which is more in line with the social phenomenon. We supplemented these contents in the introduction.
Comment 8: Materials and Methods- Please state your study design and why the design is the best according to your objectives.
Response to comment 8: We thank the reviewer for this important comment. Because human capital, social capital, and psychological capital are interdependent and have a phenomenon of ‘All roads lead to Rome’, which does not accord with the hypothesis of multiple regression. Therefore, this study used the configuration perspective to examine the relationship between the three types of career capital and well-being. The main methods used in fsQCA are Boolean algebra and set analysis, which assumes that the independent variables are interdependent and work together, and there is no optimal. Moreover, fsQCA believes that there are equivalent multiple paths or solutions (Ragin, 2000, 2014; Rihoux & Ragin, 2009; Fiss, 2011). We have supplemented this in detail in the section ‘Statistical Analysis.’
Comment 9: 3.1. Participants: inform inclusion/exclusion criteria. How and where did you find them, and how did you approach them?
Response to comment 9: We thank the reviewer for this important comment. This study set some anti-counterfeiting items, such as “Please directly select ‘fully agree.’” If the item was selected incorrectly, it would be regarded as invalid data and deleted.
Comment 10: 3.2. Materials: please be more descriptive of your measurements, including how to apply them, how long it takes, and what the main results are expected (classifications, levels, if appropriate, etc.). Please, inform the reliability and validity of these measures by citing the respective studies.
Response to comment 10: We thank the reviewer for this important comment. We have described the materials in as much detail as possible (e.g., examples of items and scoring methods). As for the questions raised by the reviewers, some questions (e.g., how long it takes and what the main results are expected) are not covered in our manuscript. Moreover, we had written the reliability and validity of each scale in the original manuscript.
Comment 11: There is no information on data collection: how long the research was available to participants, where this Survey was stored, and how they completed the measures.
Response to comment 11: We thank the reviewer for this important comment. The participants selected by convenient sampling came from 26 enterprises in 9 provinces of China. Participants completed the Likert scale by self-report. We have added these contents to section '3.1. Participants'.
Comment 12: Discussion: please include in the debate that this type of research design identifies trends; however, due to the complexity of Career Capital and Well-Being subject, it is important to conduct future qualitative studies. Qualitative studies can give you more data about “meaning”, whereas quantitative will show you trends or relations. Moreover, it is essential to address in your discussion the influence of Chinese culture on work and well-being that might differ when compared with other countries and continents.
Response to comment 12: We thank the reviewer for this important comment. FsQCA is very important for qualitative and quantitative research. Traditional case studies mainly develop theories based on the specific knowledge of a single case or a few cases and their situations. However, because they can not handle too many cases, the conclusions are not suitable for promotion. Therefore, they are often criticized by sociologists who are committed to developing general theories in comparative methods
As a new comparative analysis method, fsQCA is a combination of qualitative and quantitative methods. FsQCA replaces quantitative analysis (deviation from causal complexity) and qualitative case method (lack of generalization of research conclusions). By treating the whole case as the configuration of conditions (variables), fsQCA integrates the advantages of case study and variable study to a certain extent (Ragin, 2014). Therefore, case researchers can no longer be limited to the limitations and doubts of a few cases, and solve the causal complexity problem that cannot be solved by variable-oriented large sample analysis through overall configuration analysis.
Using QCA, researchers can also focus on the analysis of positive cases, that is, the combination of cases with only positive results. To meet the needs of theoretical saturation, traditional case studies usually require that the collected cases must have positive cases and negative cases. However, based on the asymmetry assumption of the QCA method, positive results and negative results can be the same reason, so they can be studied separately. According to the assumption of equivalence, there can be multiple combined reasons for the same result, so even if they are positive cases, we can analyze the different combined reasons for the same result. However, it should be noted that this study may be mainly a necessity analysis rather than a sufficiency analysis in the general QCA analysis.
Moreover, discussing the influence of Chinese culture on work and well-being that might differ when compared with other countries and continents is very important. Compared with the international concept of psychological capital, Chinese psychological capital emphasizes more social psychological capital, such as tolerance, forgiveness, respect, comity, modesty, honesty, gratitude, and dedication. So, the international community pays more attention to task-oriented psychological capital and neglects guanxi-oriented psychological capital. In this study, compared with guanxi-oriented psychological capital, task-oriented psychological capital is a more critical factor in achieving high well-being. Therefore, even in a collectivist society, task-oriented psychological capital is still more important. We have added these contents to the Discuss.
Thank you for your consideration. I look forward to hearing from you.
Kind regards,
Qian Xu
Round 2
Reviewer 1 Report
Accept in present form
Reviewer 2 Report
The authors reviewed the paper addressing and justifying each point of the review. I consider this paper suitable for publication.